# Feasibility of a VR Intervention to Decrease Anxiety in Children with Tumors Undergoing CVC Dressing

**DOI:** 10.3390/ijerph191911953

**Published:** 2022-09-21

**Authors:** Luisa Russo, Alberto Eugenio Tozzi, Angela Mastronuzzi, Ileana Croci, Francesco Gesualdo, Ilaria Campagna, Kiersten P. Miller, Italo Ciaralli, Matteo Amicucci, Domitilla Elena Secco, Vito Andrea Dell’Anna, Adele Ripà, Elisa Piccinelli

**Affiliations:** 1Multifactorial and Complex Diseases Research Area, Bambino Gesù Children’s Hospital, Istituto di Ricerca e Cura a Carattere Scientifico (IRCCS), 00165 Rome, Italy; 2Department of Onco Hematology and Cell and Gene Therapy, Bambino Gesù Children’s Hospital, Istituto di Ricerca e Cura a Carattere Scientifico (IRCCS), 00165 Rome, Italy

**Keywords:** virtual reality, children, cancer, CVC dressing, distress, anxiety

## Abstract

Virtual reality (VR) represents a promising digital intervention for managing distress and anxiety in children with tumors undergoing painful medical procedures. In an experimental cross-over study, we administered a VR intervention consisting of relaxing games during central venous catheter (CVC) dressing. The VR sessions were compared with no-VR during CVC medication. We used the distress thermometer and RCMAS-2 scale to assess distress and anxiety levels. We also explored the satisfaction level in patients and families. We enrolled 22 children. The distress levels after medication were lower in the VR group than in those without VR (VR: median 2; IQR 0–2; no-VR: median 4; IQR: 3–5). No variation in anxiety levels was detected by VR intervention. Satisfaction for using VR was very high in children and their families although a total of 12% of children were disappointed by the effect of VR. Most healthcare workers felt that VR would be useful in routine clinical practice. A VR intervention is highly acceptable, may be efficacious in decreasing distress in children with cancer undergoing painful procedures, but it is less likely that it has a measurable impact on anxiety. Evidence from larger studies is needed to assess VR translation into the clinical workflow.

## 1. Introduction

Children hospitalized with tumors frequently require procedures associated with anxiety and distress, which may have important effects on their and their families’ psychological well-being. Among these procedures, CVC dressing is one of the most frequent. Painful procedures may require pain control, which is usually obtained through a pharmacological intervention [1]. The standard management of invasive and potentially painful procedures in children with cancer may include different strategies, although evidence strongly supports only the use of topical anesthetics 1 h before the procedure, hypnosis, and active distraction in the management of central venous access points [2]. Virtual reality (VR) is a technology based on recreating an artificial three-dimensional interactive environment, in which the user has the illusion of movement. Several studies investigating the effect of VR during painful or distressing procedures have been conducted in adults and children [3,4]. Although VR interventions are not standardized, most available studies showed some effects on pain, with a high safety profile and high patient satisfaction [5,6]. Originally, the effect of VR on pain and distress was thought to be mainly based on distraction: pain requires attention, and VR basically works by reducing the painful stimulus through distracting the brain with another stimulus [5,7,8]. More recently, attention, emotion, memory, and other senses have been hypothesized to come into play in the analgesic effect of VR [9]. The mechanism behind the effectiveness of VR in reducing distress has been less explored in the medical literature. One study showed that surrogate nature delivered through VR elicits a restorative effect similar to that triggered by real nature settings [10].

VR has also emerged as a promising intervention to reduce pain, distress, and anxiety in patients with cancer [11,12,13]. With respect to children with hematology-oncology diseases, one study has demonstrated a realistic efficacy of VR in reducing pain and anxiety during peripherally inserted central catheter (PICC) procedures [14]. The CVC dressing change must be performed as frequently as every 7–10 days in children with cancer, often generating fear and distress. As the evidence regarding the effectiveness of VR for preventing anxiety during PICC procedures is still limited, in this study we aimed to evaluate the acceptance of VR during CVC dressing in children with tumors, to explore the effectiveness of this intervention to reduce anxiety and distress during the procedures through multiple outcome measures, and to assess the feasibility of a larger study on this subject.

## 2. Materials and Methods

This is a pilot feasibility study in which we provided an uncontrolled VR intervention to patients in the Bambino Gesù Children’s Hospital Onco-hematology Department, which includes 77 beds and a Day Hospital/outpatient clinic with 100–120 children per day. We used an experimental cross-over design, in which the effect of VR during CVC dressing change in one session was compared with another session with no VR in the same patient. The enrollment of patients has taken place over a seven-month period (from March to November 2021). We enrolled children aged 5–11-year, hospitalized in the Onco-hematology Department, who had a CVC in place, and whose parents signed an informed consent. In addition, a child consent form was submitted to all children. Exclusion criteria included patients with vision disturbances; those with impaired head movement; patients with risks from viewing digital images (e.g., photosensitive epilepsy); and patients enrolled in another research protocol which included distress/anxiety assessment.

The VR intervention consisted of a device provided by BehaVR (BehaVR, Inc., Elizabethtown, KY, USA). The equipment is a medical grade VR system including a CE-marked VR head-mounted-display (HMD) device and tablet. The HMD is a customized Pico Goblin device, with plastic straps and easily sanitizable materials. The HMD fits most head sizes and shapes. The tablet allows for clinician control and supervision of VR content via an intuitive and user-friendly interface. The HMD and the tablet are paired using Bluetooth. The VR Kit requires no internet or Wi-Fi connection. BehaVR’s VR games included:The MantaRay game, which immerses the patient in a peaceful underwater environment. Patients can use the head movement to navigate and control a manta-ray fish, which makes them get into a flow state, focusing on the present moment as they put their full attention on guiding the manta-ray. The environment is accompanied by relaxing ambient music and sound effects.The VitaminBee game is a child-friendly game that provides a source of immersive and interactive distraction. Patients need to concentrate to launch grains of pollen toward playful bees. The shooting trajectory is controlled using subtle head movements.The diaphragmatic breathing exercise immerses patients in a relaxing environment that provides audiovisual cues that entrain the patients’ breathing pace to a relaxing inhale-hold-and-exhale rhythm. The VitaminBee game was used to familiarize patients with VR, while the MantaRay game and the diaphragmatic breathing exercise were used during the CVC dressing procedure.

Before the study kick-off, healthcare workers participating in the study were trained for usage and maintenance of the BehaVR Kit.

To study the impact on distress, the VR intervention was offered to eligible children twice: (a) on a different day before the CVC dressing change to assess any effect on distress levels independently of CVC medication; and (b) during the dressing change itself. VR sessions lasted 10 min. Designated research nurses managed the administration of the VR intervention during the study and the data collection. We studied the acceptability of a VR intervention through analyzing the proportion of children and/or families that refused to participate. We then explored the changes in distress and anxiety levels comparing the sessions with VR vs no intervention.

Outcome measures included two scales for measuring distress levels and anxiety: the Distress Thermometer [15,16] and the RCMAS-2 [17]. The Pediatric Distress Thermometer consists of a self-reported scale and has been validated as a screening tool for distress in pediatric oncology [16]. A value above 3 has been indicated as reasonable for identifying distress in children [16]. The score from the RCMAS-2 scale represents the sum of the results of the whole RCMAS-2 scale and aims to detect a level of general anxiety. Scores above 60 indicate problematic levels of anxiety. To assess the consistency of responses, we also calculated the INC index. A high value of INC indicates a high probability that the subject responded inaccurately or randomly.

We also developed a satisfaction questionnaire, which was validated before the start of the study in a group of 14 children and their families. This questionnaire was administered to children and families at the end of the procedure to investigate their user experience. Finally, we interviewed the healthcare workers who participated in the study to study their perception about the applicability of VR in clinical practice or in future research studies.

We used Stata 17 for the analysis of data. We performed a descriptive statistical analysis consisting of calculation of proportions, median scores and interquartile ranges. As this was an exploratory study, we did not plan a sample size to satisfy statistical requirements. The protocol of the study has been reviewed and approved by the Ethics Committee of the Bambino Gesù Children’s Hospital (protocol code 2265_OPBG_2020 approved on 16 December 2020).

## 3. Results

Participation in the study was proposed to 24 children. Twenty-two accepted to participate. One adolescent refused the VR intervention as she preferred to directly observe the medication procedure, while the parents of a child refused the VR intervention because they deemed it as not adequate for the child to manage his anxiety.

The socio-demographic characteristics of participants are shown in Table 1. Most participants were males, their median age was 8.4 years (IQR 6.8–10.3) and nearly 70% of them had at least one sibling. The socio-demographic level of parents based on education and employment was similar to that of the general population.

Figure 1 shows the distress levels before and after the CVC dressing change in sessions with and without the VR intervention. A slight decrease of distress levels was observed in the VR group compared with no-VR in pre-medication sessions. On the other hand, the comparison of distress levels after medication by VR yielded a strong decrease in median scores. While most of the observations in the VR group after medication were below the threshold indicating distress, in all other groups the scores indicated a moderate distress.

Figure 2 shows the results obtained from the RCMAS-2 scale in the different domains. The majority of observations in RCMAS-2 scale fell in the interval between 40 and 60, indicating normal levels of anxiety. No apparent decrease was detected in the total anxiety scores and in any of the subscales with VR compared with no-VR. The inconsistency index (INC) measured in participants showed median values between 1.5 and 2 suggesting that the responses were consistent and reliable.

Figure 3 shows the perceived efficacy of VR in reducing anxiety by families and children participating in the study. A high proportion of respondents perceived VR as highly efficacious before the medication. On the other hand, VR was perceived inefficacious by 12% of children and 4% of families after medication.

The other dimensions of satisfaction collected through the questionnaire administered to families and children are illustrated in Figure 4. The vast majority of responders reported very high rates of overall satisfaction and would recommend VR to other families. Nearly 5% of families and children reported some discomfort in the use of the VR device.

Finally, regarding the opinion of healthcare workers, 7/8 of them found VR useful in clinical practice but 2/8 questioned the usability of the VR device.

## 4. Discussion

The results of this pilot study suggest that VR may reduce the distress of children undergoing CVC dressing and that this intervention is appreciated by patients, families, and healthcare professionals. A high decrease in distress levels was observed after medications with VR with distress scores mostly below the normal threshold, compared with medications without VR. Although this study was not designed for detecting statistically significant differences, the median scores of the Distress Thermometer after medication in the VR group were significantly lower than in the no-VR group.

Distress levels also decreased with VR in sessions with no medication. Stress in cancer patients may be triggered by the self-perception of the disease and the anticipated thinking of painful procedures. VR may be slightly effective when a medical practice is not imminent, and distress is experienced and perceived to a lesser extent than during the actual medication. In general, caution should be exercised when interpreting results on distress outcomes in studies on VR interventions if the baseline distress level is not taken into account. The complex combination of different determinants of distress and the expected support of the technology before medication should be also considered when interpreting results from randomized trials as blinding for a VR intervention is not feasible.

VR did not affect the anxiety levels as from the scores detected with the RCMAS-2 scale and scores were in the normal range in all the subscales. This observation is in contrast with other studies which investigated the effect of VR on anxiety in children with cancer during PICC medication [14]. We also explored the perceived efficacy with regard to anxiety of families and children before and after VR, which revealed high expectations in reducing anxiety, not always met by the results. In fact, more than 10% of children were disappointed by the VR intervention and reported no effect on anxiety.

Although our study revealed a very high feasibility of a larger experimental study on VR, we were able to detail several practical issues that should be properly managed in the design of future studies. VR intervention in children with cancer is a well accepted possibly for the novelty of the intervention and the gaming environment, and the proposal to participate in a research study encountered a negligible proportion of refusals.

Of note, the perceived usefulness of VR in clinical practice by healthcare workers was high. However, it must be noted that an extra research nurse was designated to VR administration and VR increased the medication time. This information should be taken into account when considering the translation of such a procedure into practice to avoid an increase in an already excessive workload.

The available evidence for supporting the use of VR in managing distress in children undergoing painful procedures is still scarce [18,19] and the results from some studies report inconsistent effects on distress and anxiety in children during invasive procedures [20]. Although VR has been shown sometimes effective in the management of distress and anxiety in children, the available studies on this subject largely differ for their design, intervention, and outcome measures [21,22].

Other authors have underlined the difficulties in interpreting the effect of VR in different settings and have proposed methodological recommendations to tailor the design of these studies to the experience of patients, the features of the intervention, and the available technology [23]. Our observations should help to consider a wide array of outcomes, not only to measure the impact of a VR intervention on anxiety, but to consider other specific circumstances that are essential for translating the results of a clinical study into practice.

This study has obvious limitations. Being a pilot study, we were not able to statistically assess the observed differences in the outcome measures. For the same reason, we were not able to address any difference in effect across age groups as it may be expected that the effect of VR on anxiety and distress may vary with age. On the other hand, our preliminary estimates will be useful for calculating a proper sample size for future studies aimed at investigating the efficacy of VR and stratify by age group.

## 5. Conclusions

In conclusion, although we expect that a VR intervention has a measurable impact on distress in children with cancer undergoing medications, we believe that robust evidence should rely on a comprehensive evaluation of several dimensions, including distress, anxiety, patient satisfaction, and impact on clinical workflow. As this study had only descriptive purposes, the obtained information will be useful for designing larger studies that may address more precisely the impact of VR on distress and anxiety in children undergoing stressful medical procedures.

## Figures and Tables

**Figure 1 ijerph-19-11953-f001:**
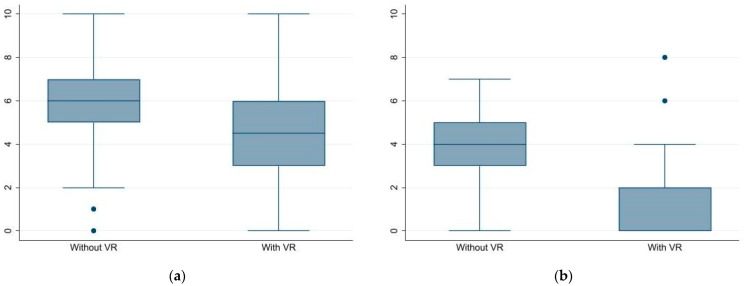
Distress Thermometer (**a**) PRE medication: without VR 6 (IQR 5–7); with VR 4.5 (IQR 3–6); (**b**) POST medication: without VR 4 (IQR 3–5); with VR 2 (IQR 0–2).

**Figure 2 ijerph-19-11953-f002:**
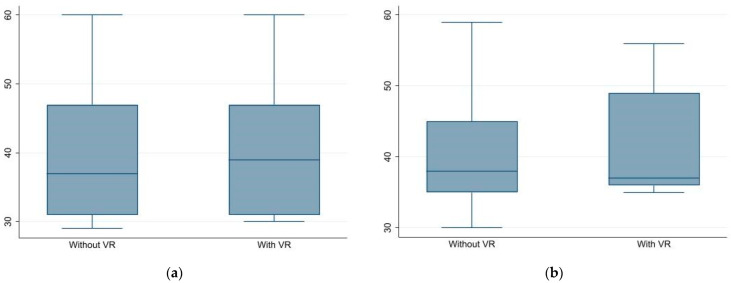
RCMAS-2 scale (**a**) physiological anxiety: without VR 37 (IQR 31–47); with VR 39 (IQR 31–47); (**b**) social anxiety: without VR 38 (IQR 35–45); with VR 37 (IQR 36–49); (**c**) concern: without VR 38.5 (IQR 31–48); with VR 37.5 (IQR 32–45); (**d**) total results: without VR 36.5 (IQR 32–48); with VR 38.5 (IQR 32–48).

**Figure 3 ijerph-19-11953-f003:**
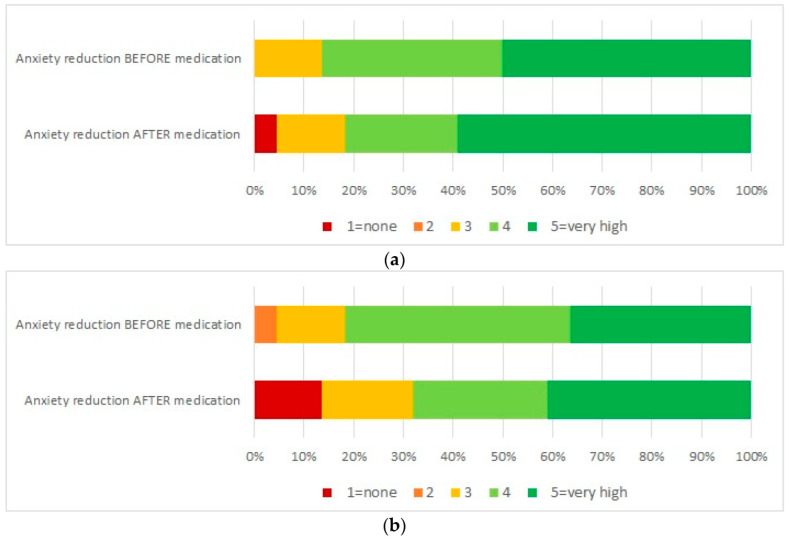
Perceived efficacy of VR by families and children (**a**) families; (**b**) children.

**Figure 4 ijerph-19-11953-f004:**
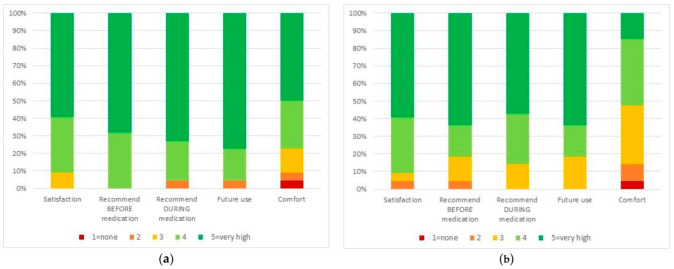
Satisfaction questionnaire results of families and children (**a**) families; (**b**) children.

**Table 1 ijerph-19-11953-t001:** Characteristics of the sample.

	N (Total = 22)
Male (%)	16 (72.7%)
Child’s age, median (IQR)	8.4 (6.8–10.3)
Mother’s age, median (IQR)	41.8 (38.4–45.4)
Mother employed (%)	12 (54.5%)
Mother with a degree (%)	8 (36.4%)
Father’s age, median (IQR)	45.3 (42.4–48.8)
Father employed (%)	22 (100.0%)
Father with a degree (%)	7 (31.8%)
At least one sibling (%)	15 (68.2%)

## Data Availability

Not applicable.

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
