# Peer review of "Feasibility of a VR Intervention to Decrease Anxiety in Children with Tumors Undergoing CVC Dressing"

_ijerph, 2022, doi:10.3390/ijerph191911953_

Round 1

Reviewer 1 Report

1. This is a pilot study with a sample size of only 24 patients which seems to be very less for concluding the fruitful results-Please justify

2. The Author did not take into consideration of other co-morbid diseases in the exclusion criteria as they can affect the results.

3. Did Author submit the Child Assent form in addition to the Guardian consent form  as it is necessary for children of the age group above 8 years to 18 yers

4. Author could have elaborated more on VR Technology used for reading anxiety, fear, and pain for a better understanding

5. Why not Author takes the study design as a Prospective Case-control study to eliminate confounding bias-please justify

6. Why not Author has taken into consideration measuring the VR effects by Functional Magnetic Resonance Imaging Technique?

Author Response

  1. This is a pilot study with a sample size of only 24 patients which seems to be very less for concluding the fruitful results-Please justify 

- Thank you very much for your considerations. The objective of the study was to explore the feasibility of a larger experimental study and inform the needed sample size through preliminary results. An exploratory study was particularly needed for evaluating the impact of VR on anxiety. Although the results lack precision due to the small sample size, we obtained important indications for planning future studies and for consolidating the study hypothesis and the study design. These limitations are reported already in the discussion and in the conclusion.

  1. The Author did not take into consideration of other co-morbid diseases in the exclusion criteria as they can affect the results. 

-Thank you so much for your consideration. All included patients did not present any comorbidities therefore in the pilot study there was no interference due to this type of problem. 

  1. Did Author submit the Child Assent form in addition to the Guardian consent form as it is necessary for children of the age group above 8 years to 18 yers 

-Thank you very much for this consideration. All children signed a child assent form before enrollment in the study. A clarification has beed added in the methods section. 

  1. Author could have elaborated more on VR Technology used for reading anxiety, fear, and pain for a better understanding 

-Thank you so much for your consideration. We expanded the information on use of VR and its rationale in the introduction.

  1. Why not Author takes the study design as a Prospective Case-control study to eliminate confounding bias-please justify 

-Thank you so much for your consideration. This is indeed a prospective experimental study. We used a cross-over design which has the advantage of making comparisons within the same individuals. As the intervention in this study could not be blinded and the sample size was limited, the cross over design was more suitable for this setting. If we had used a classical randomized clinical trial, the intervention (VR) would have been offered to only half of particpants and results would have been even more difficult to interpret.

  1. Why not Author has taken into consideration measuring the VR effects by Functional Magnetic Resonance Imaging Technique? 

-Thank you very much for this suggestion, due to limited resources available this has not been considered at this stage of the study, but it may be a valid future suggestion. A thousand thanks 

Reviewer 2 Report

Nice work, pioneristic and interesting

the very low sample size is th major limit of the study

(no conclusion can be driven ...

only suggestions)

Author Response

Nice work, pioneristic and interesting the very low sample size is th major limit of the study (no conclusion can be driven ... only suggestions) 

- Thank you so much for reviewing our manuscript and for the positive feedback. As replied to Reviewer 1, the objective of the study was to explore the feasibility of a larger experimental study and inform the needed sample size through preliminary results. An exploratory study was particularly needed for evaluating the impact of VR on anxiety. Although the results lack precision due to the small sample size, we obtained important indications for planning future studies and for consolidating the study hypothesis and the study design. These limitations are reported already in the discussion and in the conclusion.

Reviewer 3 Report

The study is really interesting, although the number of subjects is rather small. Probably a greater number of subjects will provide even more definitory results.

Author Response

The study is really interesting, although the number of subjects is rather small. Probably a greater number of subjects will provide even more definitory results. 

-Thank you very much for reading and revising our manuscript and thanks for the appreciation. We look forward to publishing the results of the larger study later.